# P2RBox: A Single Point is All You Need for Oriented Object Detection

## Abstract

Oriented object detection, a specialized subfield in computer vision, finds applications across diverse scenarios, excelling particularly when dealing with objects of arbitrary orientations. Conversely, point annotation, which treats objects as single points, offers a cost-effective alternative to rotated and horizontal bounding boxes but sacrifices performance due to the loss of size and orientation information. In this study, we introduce the P2RBox network, which leverages point annotations and a mask generator to create mask proposals, followed by filtration through our Inspector Module and Constrainer Module. This process selects high-quality masks, which are subsequently converted into rotated box annotations for training a fully supervised detector. Specifically, we've thoughtfully crafted an Inspector Module rooted in multi-instance learning principles to evaluate the semantic score of masks. We've also proposed a more robust mask quality assessment in conjunction with the Constrainer Module. Furthermore, we've introduced a Symmetry Axis Estimation (SAE) Module inspired by the spectral theorem for symmetric matrices to transform the top-performing mask proposal into rotated bounding boxes. P2RBox performs well with three fully supervised rotated object detectors: RetinaNet, Rotated FCOS, and Oriented R-CNN. By combining with Oriented R-CNN, P2RBox achieves 62.26% on DOTA-v1.0 test dataset. As far as we know, this is the first attempt at training an oriented object detector with point supervision.

## 1 Introduction

Aerial object detection focuses on identifying objects of interest, such as vehicles and airplanes, on the ground within aerial images and determining their categories. With the increasing availability of aerial imagery, this field has become a specific yet highly active area within computer vision (Ren et al., 2015; Lin et al., 2017; Tian et al., 2019; Ding et al., 2019; Xie et al., 2021).

Nevertheless, obtaining high-quality bounding box annotations demands significant human resources. Weakly supervised object detection (Bilen & Vedaldi, 2016; Tang et al., 2017; 2018; Chen et al., 2020; Wan et al., 2018; Zhou et al., 2016; Diba et al., 2017; Zhang et al., 2018) has emerged as a solution, replacing bounding box annotations with more affordable image-level annotations. However, due to the absence of precise location information and challenges in distinguishing densely packed objects, image-level supervised methods exhibit limited performance in complex scenarios. In recent times, point-based annotations have gained widespread usage across various tasks, including object detection (Papadopoulos et al., 2017; Ren et al., 2020), localization (Yu et al., 2022; Ribera et al., 2019; Song et al., 2021), instance segmentation (Cheng et al., 2022), and action localization (Lee & Byun, 2021).

One intriguing question naturally arises: Can weakly supervised learning for oriented object detection be achieved solely using point annotations instead of rotated bounding box annotations? We explore this question using a mask proposal generator (*e.g.*, SAM (Kirillov et al., 2023) as employed in this paper). One straightforward approach is to choose the mask with the highest associated score as the object. Following this, we apply the minimum bounding rectangle method to transform it into rotated bounding box annotations, which serves as our baseline.

However, due to the lack of intra-class homogeneity, ambiguity arises between companion scores and the best-performing mask (the one with the highest Intersection over Union with the ground

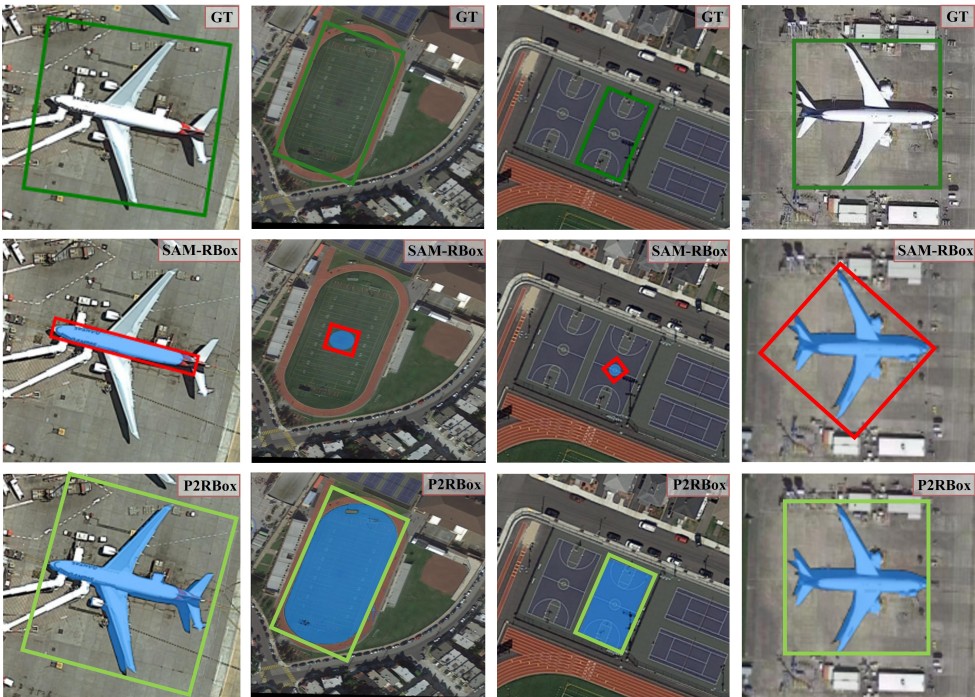

Figure 1: Visual comparison of the highest confidence mask and its corresponding rotated box annotation generated by mask proposal generator (SAM) and P2RBox. The second row displays the results of the baseline method, while the Rotated boxes in the last row are generated by the SAE module. (Best viewed in color).

truth). This ambiguity leads to difficulties in selecting the best-performing mask based on companion scores, as illustrated in Fig. 1. In this paper, we introduce an architecture inspired by Multiple Instance Learning (MIL). By extracting mask proposal information during the Inspector Module, the network becomes proficient in classifying specific objects by aggregating information from annotated points across the entire dataset. This results in a semantic score that enhances the assessment of proposal masks. Additionally, we introduce the Constrainer Module, which takes into account the alignment between marked points and the center of gravity of a mask, providing offset penalties. After aggregating all these assessments, the best mask is selected using the new assessment criteria, and it is used to generate a rotated circumscribed bounding box via the Symmetry Axis Estimation (SAE) Module, as illustrated in Fig. 2. Our main contributions are as follows:

**1)** Proposing of the P2RBox Network: We introduce the P2RBox network, which is a method based on point annotation and a mask generator for achieving point-supervised rotated object detection. To the best of our knowledge, this marks the first attempt to train a rotated object detector using point supervision. By combining with Oriented R-CNN, P2RBox achieves 62.26% on DOTA-v1.0 test dataset.

**2)** High-Quality Mask Selection: Utilizing the Inspector Module, we introduce a semantic score for the masks, combined with the Constrainer Module, leading to the development of a comprehensive filtering approach. This, in turn, enhances the quality of the selected mask proposals from the mask generator and ultimately improves the quality of rotated bounding box annotations.

**3)** Mask-to-Rotated Box Conversion: We design the SAE module based on the spectral theorem for symmetric matrices to convert the best mask proposal into rotated bounding boxes, enhancing the effectiveness of rotated object detection.

## 2 RELATED WORK

**RBox-supervised Oriented Object Detection.** Notable approaches in this field include Rotated RetinaNet (Lin et al., 2017) with anchor-based methods, Rotated FCOS (Tian et al., 2019) using anchor-free techniques, and two-stage detectors like RoI Transformer (Ding et al., 2019), Oriented

R-CNN (Xie et al., 2021), and ReDet (Han et al., 2021b). Performance enhancements have been seen with methods like R3Det (Yang et al., 2021b) and S2A-Net (Han et al., 2021a), leveraging alignment features. Most of these approaches use direct angle regression, but this can face challenges due to the periodic nature of angles, leading to strategies like modulated losses (Yang et al., 2019a; Qian et al., 2021), angle coders (Yang & Yan, 2020; Yang et al., 2021a; Yu & Da, 2023), and Gaussian-based losses (Yang et al., 2021c;d; 2022b;c). RepPoint-based approaches (Yang et al., 2019b; Hou et al., 2023; Li et al., 2022) provide alternative solutions for oriented object detection by predicting a set of sample points defining the object's spatial extent.

**HBox-supervised oriented object detection.** While the oriented bounding box can be derived from the segmentation mask, employing the HBox-Mask-RBox pipeline can be less efficient in terms of cost. A pioneering approach, H2RBox (Yang et al., 2022a), bypasses the segmentation step and directly detects RBoxes from HBox annotations. By leveraging HBox annotations for the same object in various orientations, the geometric constraints narrow down the possible angles for the object, making the detection more efficient. Additionally, the integration of a self-supervised branch in H2RBox helps filter out undesirable results, establishing an HBox-to-RBox paradigm. In an extended version, H2RBox-v2 (Yu et al., 2023), a new self-supervised branch further enhances the precision of object angle learning, resulting in improved performance.

**Point-supervised object detection.** Point-level annotation, a recent advancement, is efficient, taking about 1.87 seconds per image on the VOC dataset (Everingham et al., 2010), comparable to image-level annotation (1.5 seconds per image) and much less than bounding box annotation (34.5 seconds per image), especially rotated bounding box annotation. However, the time for point-level annotation may increase with more objects in the image. P2BNet (Chen et al., 2022) uses a coarse-to-fine strategy, enhancing IoU with ground-truth by predicting pseudo boxes using point annotations and Faster R-CNN (Ren et al., 2015).

In a related context, (Yu et al., 2022) explores object localization with coarse point annotations, addressing point annotation's semantic variability through algorithms. Additionally, (He et al., 2023) predicts horizontal bounding boxes in remote sensing scenes using point annotations. The Segment Anything Model (SAM) (Kirillov et al., 2023) allows obtaining object masks with a simple click, but ensuring mask quality remains challenging. Combining P2BNet (Chen et al., 2022) and H2RBox-v2 (Yu et al., 2023) provides the final object orientation, but H2RBox-v2 requires precise circumscribed horizontal bounding boxes. Background noise may affect both P2BNet and H2RBox-v2, leading to a poor performance (line of P2BNet-H2RBox in Tab. 1). Therefore, P2RBox focuses on generating high-quality rotated bounding boxes, a domain that has yet to be extensively studied to date.

## 3 POINT-TO-ROTATED-BOX NETWORK

As shown in Fig. 2, we design the P2RBox to establish a seamless connection between point annotations and rotated boxes through the generation, constraint, and inspection of mask proposals. Specifically, the annotated point located on an object serves as the prompt for producing initial mask proposals. Subsequently, a dedicated Constrainer Module is devised to discern and refine the most plausible masks. Building upon these filtered candidates, we introduce a novel Inspector Module designed to discern superior masks by intuitively capturing the semantic nuances embedded within the masks. Lastly, our improved mask-to-oriented-box module, named SAE, plays a pivotal component in facilitating the annotation transformation.

### 3.1 CONSTRAINER MODULE

In many cases, the annotated point of an object is typically positioned in close proximity to the center of the mask (Chen et al., 2022). Leveraging this observation, we introduce a penalty term that quantifies the distance between the mask's center and the annotated point, thus facilitating the prioritization of high-quality masks while effectively filtering out masks with excessive background content.

**Centroid Offset Penalty.** Let $Radius$ represent the farthest Euclidean distance from any pixel on the mask to the annotation point, and $dis$ denote the offset of the pixel's centroid on the mask from

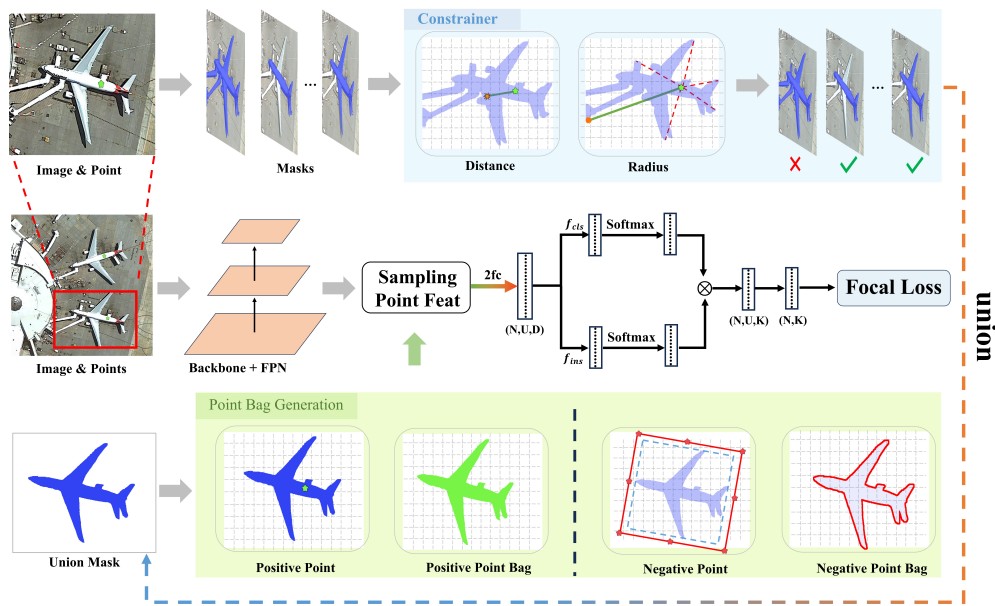

Figure 2: The overview of training process of P2RBox, consisting of mask generator, Constrainer Module and Inspector Module. Initially, mask proposals are generated by a generator (SAM). The Constrainer Module selects high-quality masks to create the union mask. Four point sets are constructed according the union mask to train Inspector Moudle, which pursuing dataset-wide category consistency. The trained network will used to assesses mask quality, selecting the best proposals for detector training supervision. (Best viewed in color)

the annotation point. A penalty formula regarding the relative offset is designed as follows:

$$S_{offset} = (1 - \exp(-w \cdot Radius + b)) \cdot \frac{dis}{Radius}. \tag{1}$$

While $dis$ is scale-sensitive, we use $dis/Radius$ to establish a scale-independent criterion. This approach works well for larger objects. However, consider small objects, we incorporate the term $1 - \exp(-w \cdot Radius + b)$ to increase tolerance. This adjustment is necessary because even a single-pixel offset during annotation can result in significant changes. The term $1 - \exp(-w \cdot Radius + b)$ ($w$ set as 0.04, $b$ set as 0.01, spectively) indicates that the coefficient is positively associated with the Radius. This suggests that the Constrainer operates under the assumption that as the $dis/Radius$ ratio remains constant, a reduction in the value of $Radius$ results in a corresponding decrease in $S_{offset}$. In our network, we only retain masks for which $S_{offset}$ is less than the threshold $thr1$, set to 0.15 in our paper. During training, all masks meeting the criteria mentioned will be merged into a unified mask for subsequent steps.

## 3.2 INSPECTOR MODULE

Utilizing the qualified masks derived from the Constrainer Module, the Inspector Module samples positive and negative points to guide the model in acquiring deeper perception of the semantic category associated with specific objects, thereby enhancing the assessment of the proposal masks.

**Point Sampling.** The four points bag or set (positive bags, negative bags, negative points and annotation points.) are constructed to train inspector module, which are described here. We denote the coordinates of an annotated point as $a \in \mathbb{R}^2$ and its corresponding category as $c \in \{0, 1\}^K$, where $K$ represents the total number of categories. $p = (p_x, p_y)$ denotes a point on a feature map.

**1) Positive Bag Construction.** In Fig. 2, with a relatively trustworthy mask in the neighborhood of $a$, we define $Radius$ as the maximum Euclidean distance between the annotated point and pixels forming the mask. We define $N$ ring-shaped sampling regions. Then we randomly sample $u_0$ points within each region, and obtain $Sample(a, r)$. All sampled points of $N$ ring-shaped regions are

defined as points' bag of $a$, denoted as $\mathcal{B}$ in Eq. 2.

$$Ring(a, r) = \left\{p|p \in mask, \frac{r-1}{N} < \frac{||p-a||}{Radius} <= \frac{r}{N}\right\}, 1 <= r <= N. \tag{2}$$

$$Sample(a, r) = \{p_i|p_i \in Ring(a, r), \ p_i \ is \ randomly \ selected\}. \tag{3}$$

$$\mathcal{B} = \bigcup_{1 \leq r \leq N} Sample(a, r), \tag{4}$$

where $\mathcal{B}$ is used for calculating the MIL loss for P2RBox training, and $|Sample(a, r)| = u_0$ ($u_0$ are number of points sampled for each ring).

**2) Negative Bag Construction.** Background points located far from the object are easier for the network to learn. Conversely, the negative points along the object's boundary are more valuable samples, as they often reside on the critical boundary between the foreground and background. Hence, we design a negative point bag, training the model to discern boundary features. By selecting $u_1$ points in the margin pixels of the mask, the negative point bag of $a_j$ can be defined as:

$$\mathcal{B}_{neg} = \left\{p_i|p_i \in mask_{margin}, \ p_i \ is \ randomly \ selected\right\}, \tag{5}$$

where $|\mathcal{B}_{neg}| = u_1$ and $mask_{margin}$ can be obtained by calculating the non-zero points on the gradient map (implemented using first-order differences) of the mask.

**3) Negative Points.** With annotated point $a$ as a naturally positive reference, we introduce negative points to instruct the model in discerning background features. To obtain the negative points for a given mask, we follow a three-step process. Firstly, determine the circumscribed bounding box of the mask, denoted as $(x, y, w, h, \alpha)$. Secondly, increase both the height $h$ and width $w$ by a factor as $\hat{h} = (1 + \delta) \cdot h$ and $\hat{w} = (1 + \delta) \cdot w$, where $\delta$ is the set to $0.05$ in this paper. Lastly, the set of negative points, denoted as $\mathcal{N}$, comprises the four vertices along with the midpoints of the four edges, $i.e.$,

$$n_{ij} = (x + \frac{\hat{w}}{2} \cdot \cos\alpha \cdot i - \frac{\hat{h}}{2} \cdot \sin\alpha \cdot j, \ y + \frac{\hat{w}}{2} \cdot \sin\alpha \cdot i + \frac{\hat{h}}{2} \cdot \cos\alpha \cdot j),$$
$$\mathcal{N} = \left\{n_{ij} \,|\, i, j \in \{-1, 0, 1\}, (i, j) \neq (0, 0)\right\}. \tag{6}$$

**P2RBox Loss.** In following, we gives the details of the objective function of training P2RBox network based on the positive bag $\mathcal{B}$ and its score $S_{\mathcal{B}}$, negative bag $\mathcal{B}_{neg}$ and its score $S_{\mathcal{B}_{neg}}$, and negative points $\mathcal{N}$. Based on the designed loss as described, after training, the network has acquired pixel-level classification capability.

**1) Bag Score.** To facilitate P2RBox in determining whether the points in $\mathcal{B}$ belong to the same category as $a$, we treat the points in bag $\mathcal{B}$ as positive instances. We extract the feature vectors $\{\mathbf{F}_p|p \in \mathcal{B}\}$. For each $p \in \mathcal{B}$, two separate fully connected layers with independent weights generate the classification score and instance score, denoted as $[S_{\mathcal{B}}^{ins}]_p$ and $[S_{\mathcal{B}}^{cls}]_p$.

To obtain the final classification score for $\mathcal{B}$, we calculate it by summing the element-wise products of $[S_{\mathcal{B}}^{ins}]_p$ and $[S_{\mathcal{B}}^{cls}]_p$ as follows:

$$S_{\mathcal{B}} = \sum_{p \in \mathcal{B}}[S_{\mathcal{B}}^{ins}]_p \cdot [S_{\mathcal{B}}^{cls}]_p, \in \mathbb{R}^K. \tag{7}$$

This score will be used for the subsequent loss calculation.

**2) Total Loss.** Object-level MIL loss is introduced to endow P2RBox the ability of finding semantic points around each annotated point. By combining information from similar objects throughout the entire dataset, it imparts discriminative capabilities to the features of that category, distinguishing between foreground and background. The objective function of P2RBox is a weighted summation of the three losses:

$$\mathcal{L}_{P2RBox} = \mathcal{L}_{ann} + \mathcal{L}_{MIL}^{pos} + \mathcal{L}_{MIL}^{neg} + \mathcal{L}_{neg}. \tag{8}$$

And $\mathcal{L}_{MIL}, \mathcal{L}_{ann}$ and $\mathcal{L}_{neg}$ are based on the focal loss (Lin et al., 2017):

$$FL(S_p, c) = \sum_{k=1}^{K} c_k(1 - S_{p,k})^\gamma \log(S_{p,k}) + (1 - c_k)S_{p,k}^\gamma \log(1 - S_{p,k}), \tag{9}$$

where $\gamma$ is set as 2 following the standard focal loss. $S_p \in \mathbb{R}^K$ and $c \in \{0,1\}^K$ are the predicted scores on all categories and the category label, respectively.

**3) Object-level MIL Loss.** As mentioned above, $S_{\mathcal{B}}$ is computed for a set of points in bag $\mathcal{B}$. Each point's classification score and instance score are generated independently. The final score is obtained by summing the element-wise products of these scores. Based on bag score $S_{\mathcal{B}}$, the MIL loss is given by the focal loss with the category label $c$ of $a$:

$$\mathcal{L}_{MIL}^{pos} = \frac{1}{M} \sum_{j=1}^{M} \mathrm{FL}(S_{\mathcal{B}_j}, c_j). \tag{10}$$

In a similar manner, $\mathcal{L}_{MIL}^{neg}$ can be computed by applying the same procedure to $\mathcal{B}_{neg}$.

**4) Annotaion Loss.** Due to the absence of positive samples with point annotations, the annotated points serve as natural positive samples that guide the model in learning about the foreground of each category. Hence, we introduce the annotation loss $L_{ann}$ to provide the network with accurate positive samples for supervision. $L_{ann}$ ensures a high score for annotated points. A classification branch with shared weights, as described above, is utilized to compute $L_{ann}$ in the following manner.

$$S_a = \sigma(fc^{cls}(\mathbf{F}_a)) \in \mathbb{R}^K,$$
$$\mathcal{L}_{ann} = \frac{1}{M} \sum_{j=1}^{M} \mathrm{FL}(S_{a_j}, c_j), \tag{11}$$

where $M$ is the number of objects in an image, $\sigma$ served as an activation function.

**5) Negative Loss.** Conventional MIL employs binary logarithmic loss and considers proposals from other categories as negative samples. Due to the absence of explicit supervision from background samples, it struggles to effectively suppress the negative samples during MIL training. To address this, we calculate the negative loss, denoted as $L_{neg}$, which constitutes the negative component of the focal loss. The calculation is as follows, with $\gamma$ set to 2, based on the set $N_j$.

$$S_p = \sigma_1(fc^{cls}(\mathbf{F}_p)) \in \mathbb{R}^K;$$
$$\mathcal{L}_{neg} = \frac{1}{8 * M} \sum_{j=1}^{M} \sum_{p \in N_j} S_p^{\gamma} \cdot \log(1 - S_p). \tag{12}$$

**Mask Quality Assessment.** By assimilating location and category information from annotated points, the Inspector Module gains classification capability. This allows for the prediction of semantic information, enhancing its quality assessment of the mask. Additionally, the classification scores of the marginal points (*i.e.*, negative bag) are taken into account and integrated to derive the semantic score of the mask:

$$S_{smt} = \alpha_1 \cdot mask^{cls} - \alpha_2 \cdot mask_{margin}^{cls}, \tag{13}$$

where, $mask^{cls}$ represents the mean of classification scores across all pixels within the mask that pertain to the identical class as the annotated point. Similarly, the computation of $mask_{margin}$ follows a akin approach.

We enhance the mask selection process by incorporating mask-associated scores with the center of mass deviation penalty introduced by the Constrainer Module. This results in a comprehensive weighted average score, derived from three quantified scores, which surpasses the performance achieved by using mask-associated scores alone.

$$Score = S_{mask} - \beta_1 \cdot S_{offset} + \beta_2 \cdot S_{smt}, \tag{14}$$

where $S_{mask}$ is accompanied by its inherent properties at the moment of its generation given by SAM, $S_{offset}$ is defined in Constrainer Module.

During the testing phase, we straightforwardly select the mask with the highest score as the object's mask, which is subsequently converted into a rotated bounding box.

### 3.3 SYMMETRY AXIS ESTIMATION.

Symmetry Axis Estimation (SAE) Module primarily addresses a specific yet prevalent issue. In the case of a symmetrical object, its symmetry axes are typically considered as its orientation. As we are aware, rotated bounding boxes offer a more precise means of annotation and can also convey its orientation. Generally, an object's orientation aligns with at least one edge of the minimum circumscribed rectangle. However, this isn't always the case. For instance, consider an object like a plane; even though it possesses an axis of symmetry, its smallest enclosing rectangle does not have any edges parallel to its orientation, as shown in the last column of Fig. 1.

**Symmetry Axis Direction.** Assuming the presence of a symmetric object, let's denote all its pixel coordinates as $P$, which forms an $n \times 2$ matrix. By translating its center of mass to the origin, we ensure that the origin always coincides with its symmetric axis. We assert that *the eigenvectors of the matrix $P^T \cdot P$ correspond to the object's symmetry and vertical directions.*

**Proof of The Assertion.** In accordance with this condition, if the target exhibits symmetry along an axis passing through the origin, then there exists a rotation matrix, also referred to as an orthogonal matrix $R$, such that:

$$P \cdot E = Q \cdot R, E = \begin{pmatrix} 1 & 0 \\ 0 & 1 \end{pmatrix}. \tag{15}$$

Here, we have a matrix $R$ with dimensions $2 \times 2$, representing a rotated matrix that aligns the axis of symmetry with the x-axis. Consequently, we can express $Q$ as follows:

$$Q = \begin{pmatrix} x_1 & x_1 & \dots & x_n & x_n \\ y_1 & -y_1 & \dots & y_n & -y_n \end{pmatrix}^T. \tag{16}$$

To find the matrix $R$, we multiply both sides of the above equation by its transpose, yielding:

$$E^T \cdot P^T \cdot P \cdot E = R^T \cdot Q^T \cdot Q \cdot R. \tag{17}$$

This further simplifies to:

$$P^T \cdot P = R^T \cdot \begin{pmatrix} 2 \times \sum_{i=1}^{n} x_i^2 & 0 \\ 0 & 2 \times \sum_{i=1}^{n} y_i^2 \end{pmatrix} \cdot R. \tag{18}$$

By spectral theorem for symmetric matrices, Eq. 18 demonstrates that $Q^T \cdot Q$ and $P^T \cdot P$ are similar matrices, with $R$ serving as the similarity transformation matrix and also the eigenvector matrix of $P^T \cdot P$ because $Q^T \cdot Q$ being a diagonal matrix. This confirms our assertion: The eigenvectors of the matrix $P^T \cdot P$ correspond to the object's symmetry direction and its vertical direction.

In the SAE Module, we generate oriented bounding rectangles for categories PL and HC, following their symmetry axes. For simplicity, other categories continue to use minimum circumscribed bounding boxes.

## 4 EXPERIMENTS

To evaluate our proposed method, we conduct extensive experiments on the most widely-used oriented object detection datasets, namely DOTA (Xia et al., 2018).

### 4.1 DATASETS AND IMPLEMENT DETAILS

**DOTA.** There are 2,806 aerial images—1,411 for training, 937 for validation, and 458 for testing, as annotated using 15 categories with 188,282 instances in total. We follow the preprocessing in MMRotate—The high-resolution images are split into $1,024 \times 1,024$ patches with an overlap of 200 pixels for training, and the detection results of all patches are merged to evaluate the performance. We use training and validation sets for training and the test set for testing. The detection performance is obtained by submitting testing results to DOTA's evaluation server. We report the $AP_{50}$ which uses the IoU between the predicted rotated boxes and rotated ground-truth bounding boxes.

**Training Details.** P2RBox predicts the rotated bounding boxes from single point annotations and uses the predicted boxes to train three classic oriented detectors (RetinaNet, FCOS, Oriented R-CNN) with standard settings. All the fully-supervised models are trained based on a single GeForce

RTX 2080Ti GPU. Our model is trained with SGD (Bottou, 2012) on a single GeForce RTX 2080Ti GPU. The initial learning rate is $2.5 \times 10^{-3}$ with momentum 0.9 and weight decay being 0.0001. And the learning rate will warm up for 500 iterations.

## 4.2 MAIN RESULT

Table 1: Results of each category on the DOTA-v1.0 test set.

| Method | PL | BD | BR | GTF | SV | LV | SH | TC | BC | ST | SBF | RA | HA | SP | HC | mAP$_{50}$ |
|---|---|---|---|---|---|---|---|---|---|---|---|---|---|---|---|---|
| *Rbox-supervised:* | | | | | | | | | | | | | | | | |
| RetinaNet (2017) | 89.1 | 74.5 | 44.7 | 72.2 | 71.8 | 63.6 | 74.9 | 90.8 | 78.7 | 80.6 | 50.5 | 59.2 | 62.9 | 64.4 | 39.7 | 67.83 |
| FOCS (2019) | 88.4 | 75.6 | 48.0 | 60.1 | 79.8 | 77.8 | 86.7 | 90.1 | 78.2 | 85.0 | 52.8 | 66.3 | 64.5 | 68.3 | 40.3 | 70.78 |
| Oriented R-CNN (2021) | 89.5 | 82.1 | 54.8 | 70.9 | 78.9 | 83.0 | 88.2 | 90.9 | 87.5 | 84.7 | 64.0 | 67.7 | 74.9 | 68.8 | 52.3 | 75.87 |
| RepPoints (2019b) | 84.8 | 73.4 | 40.7 | 56.5 | 71.6 | 52.2 | 73.4 | 90.6 | 76.3 | 85.2 | 58.8 | 61.4 | 54.9 | 64.4 | 18.6 | 64.18 |
| Faster R-CNN (2015) | 88.4 | 73.1 | 44.9 | 59.1 | 73.3 | 71.5 | 77.1 | 90.8 | 78.9 | 83.9 | 48.6 | 63.0 | 62.2 | 64.9 | 56.2 | 69.05 |
| RoI Transformer (2019) | 88.6 | 78.5 | 43.4 | 75.9 | 68.8 | 73.7 | 83.6 | 90.7 | 77.3 | 81.5 | 58.4 | 53.5 | 62.8 | 58.9 | 47.7 | 69.56 |
| DAL (2021) | 88.7 | 76.6 | 45.1 | 66.8 | 67.0 | 76.8 | 79.7 | 90.8 | 79.5 | 78.5 | 57.7 | 62.3 | 69.1 | 73.1 | 60.1 | 71.44 |
| RSDet (2021) | 89.8 | 82.9 | 48.6 | 65.2 | 69.5 | 70.1 | 70.2 | 90.5 | 85.6 | 83.4 | 62.5 | 63.9 | 65.6 | 67.2 | 68.0 | 72.20 |
| R$^3$Det (2021b) | 88.8 | 83.1 | 50.9 | 67.3 | 76.2 | 80.4 | 86.7 | 90.8 | 84.7 | 83.2 | 62.0 | 61.4 | 66.9 | 70.6 | 53.9 | 73.79 |
| *Hbox-supervised:* | | | | | | | | | | | | | | | | |
| BoxInst-RBox (2021) | 68.4 | 40.8 | 33.1 | 32.3 | 46.9 | 55.4 | 56.6 | 79.5 | 66.8 | 82.1 | 41.2 | 52.8 | 52.8 | 65.0 | 30.0 | 53.59 |
| H2RBox (2022a) | 88.5 | 73.5 | 40.8 | 56.9 | 77.5 | 65.4 | 77.9 | 90.9 | 83.2 | 85.3 | 55.3 | 62.9 | 52.4 | 63.6 | 43.3 | 67.82 |
| H2RBox-v2 (2023) | 89.0 | 74.4 | 50.0 | 60.5 | 79.8 | 75.3 | 86.9 | 90.9 | 85.1 | 85.0 | 59.2 | 63.2 | 65.2 | 70.5 | 49.7 | 72.31 |
| *Point-supervised:* | | | | | | | | | | | | | | | | |
| P2BNet-H2RBox | 2.3 | 33.8 | 1.2 | 3.6 | 36.7 | 10.2 | 22.3 | 0.2 | 1.6 | 24.5 | 9.1 | 44.4 | 10.5 | 34.8 | 20.9 | 17.08 |
| SAM (RetinaNet) | 79.7 | 64.6 | 11.1 | 45.6 | 67.9 | 47.7 | 74.6 | 81.1 | 6.6 | 75.7 | 20.0 | 30.6 | 36.9 | 50.5 | 26.1 | 47.91 |
| SAM (FCOS) | 78.2 | 61.7 | 11.7 | 45.1 | 68.7 | 64.8 | 78.6 | 80.9 | 5.0 | 77.0 | 16.1 | 31.8 | 45.7 | 53.4 | 44.2 | 50.84 |
| SAM (Oriented R-CNN) | 79.0 | 62.6 | 8.6 | 55.8 | 68.4 | 67.3 | 77.2 | 79.5 | 4.4 | 77.1 | 26.9 | 28.8 | 49.2 | 55.2 | 51.3 | 52.75 |
| **P2RBox** (RetinaNet) | 86.9 | 70.0 | 12.5 | 47.9 | 70.4 | 53.9 | 75.4 | 88.8 | 44.1 | 77.4 | 41.9 | 33.4 | 41.2 | 53.9 | 34.8 | 55.50 |
| **P2RBox** (FCOS) | 86.7 | 66.0 | 14.5 | 47.4 | 72.4 | 71.3 | 78.6 | 89.7 | 45.8 | 79.6 | 44.6 | 34.8 | 48.4 | 55.4 | 40.8 | 58.40 |
| **P2RBox** (Oriented R-CNN) | 87.7 | 72.6 | 13.9 | 63.1 | 70.1 | 74.7 | 82.8 | 90.1 | 46.4 | 81.8 | 53.0 | 33.5 | 57.2 | 56.4 | 50.1 | 62.26 |

As shown in Tab.1, our model's performance across many categories is astonishing. In point-supervised detection, to demonstrate the effectiveness of the proposed method in our model, we designed a parameter-free rotation box annotation generator based on SAM, which directly retains the highest-score mask and computes the minimum bounding rectangle to obtain the rotated bounding box. By comparing the results of pseudo-label training on three strong supervised detectors, P2RBox model outperforms our baseline in every single category combined with any detector (55.50% vs. 47.91% on RetinaNet, 58.40% vs. 50.84% on FCOS, 62.26% vs. 52.75% on Oriented R-CNN).

Our mAP$_{50}$ is 62.26% combined with Oriented R-CNN, which exceeds the previous methods with the H2Rbox-based detector, e.g., BoxInst-RBox (Tian et al., 2021). Compared with the H2Rbox, P2RBox (Oriented R-CNN) achieves comparable performance in some categories, such as GTF and HC. Examples of detection results on the DOTA dataset using P2RBox (Oriented R-CNN) are shown in Fig. 3

## 4.3 ABLATION STUDY

The ablation study's mAP results are based on P2RBox (RetinaNet), mIoU results are calculated between ground truth and pseudo rotated box. The following experiments assume that $\alpha_1 = \alpha_2 = \beta_1 = \beta_2 = 1.0$ and the SAE method is applied on PL and HC if not specified.

**P2RBox Modules.** As depicted in Tab. 2, we evaluated various strategies of the P2RBox model, including Inspector Module, Constrainer Module, and the Symmetry Axis Estimation Module. Our experiments reveal the following: The first row of Tab. 2 indicates that we have confidence in the score provided by SAM for selecting the highest-scoring mask proposal. Then, we transform this mask proposal into a rotated box by calculating its minimum bounding rectangle, resulting in an mAP of 47.91%. Subsequent experiments demonstrate the performance improvements achieved by each module.

**Assessment Score.** Our mask assessment score consists of three weighted quantified scores in Eq. 14. The weight parameter $\beta_1, \beta_2$ is used to get the final score for selecting. As shown in Tab. 3, our model demonstrates insensitivity to parameter adjustments, showing robustness.

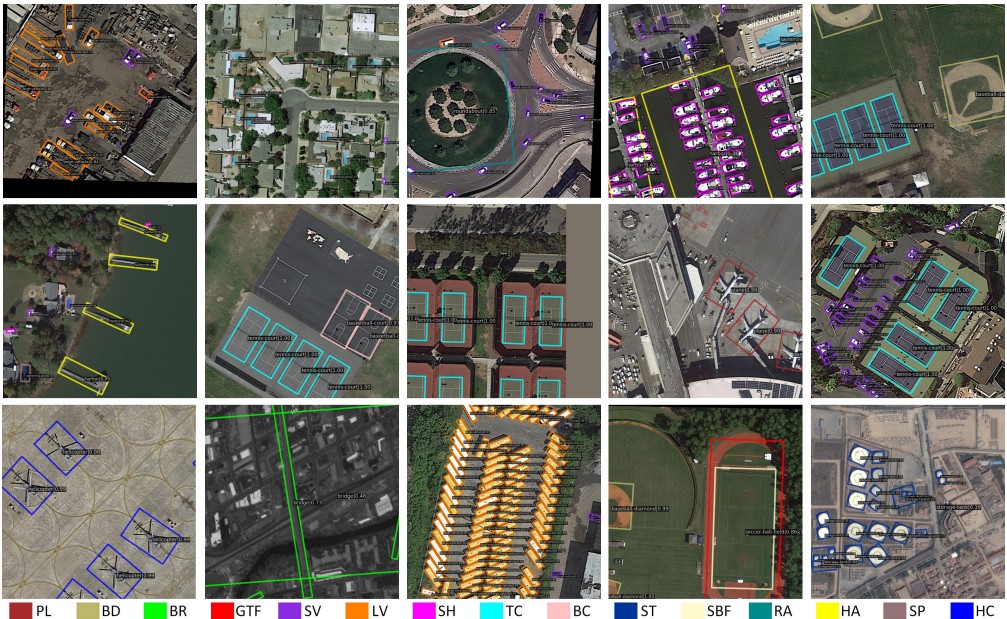

| PL | BD | BR | GTF | SV | LV | SH | TC | BC | ST | SBF | RA | HA | SP | HC |

Figure 3: Examples of detection results on the DOTA dataset using P2RBox (Oriented R-CNN).

**Semantic Score.** In the Inspector Module, to harness the full potential of the network's semantic capabilities, we conducted experiments concerning the parameters $\alpha_1$ and $\alpha_2$. As demonstrated in Tab. 4, when we set $\alpha_2 = 0$, the mIoU decreases from 60.68% to 57.90%, when set $\alpha_1 = 0$, the mIoU decreases from 60.68% to 52.74%. This indicates that the model possesses the ability to distinguish between the margin and inner pixels. The subsequent experiments demonstrate that the mIoU is not significantly affected, and it is acceptable to set $\alpha_1 = \alpha_2 = 1.0$.

**SAE Method.** To show the full performance of SAE method, in major categories. As shown in Tab. 5, our SAE method exhibits significant improvements compared with minimum bounding rectangle in both plane (PL) and helicopter (HC), while keeping other categories basically the same.

Table 2: Ablation with main modules. Ins for Inspector, Cons for Constrainer.

Table 3: Varying $\beta_1$, $\beta_2$ for assessment score.

Table 4: Varying $\alpha_1$, $\alpha_2$ for semantic score.

| Ins | Cons | SAE | mIoU | mAP |
|---|---|---|---|---|
| - | - | - | 54.86 | 47.91 |
| ✓ | - | - | 55.98 | 49.33 |
| - | ✓ | - | 58.77 | 53.49 |
| ✓ | ✓ | - | 59.68 | 54.38 |
| ✓ | ✓ | ✓ | 60.68 | 55.50 |

| $\beta_1$ | $\beta_2$ | mIoU |
|---|---|---|
| 1.2 | 1.2 | 60.69 |
| 1 | 1 | 60.68 |
| 0.8 | 0.8 | 60.64 |
| 1.2 | 0.8 | 60.62 |
| 0.8 | 1.2 | 60.13 |

| $\alpha_1$ | $\alpha_2$ | mIoU |
|---|---|---|
| 1.2 | 1.2 | 60.44 |
| 1 | 1 | 60.68 |
| 0.8 | 0.8 | 60.87 |
| 1.2 | 0.8 | 60.49 |
| 0.8 | 1.2 | 59.29 |

Table 5: IoU results in major categories on DOTA using different methods.

| Method | PL | BR | SV | LV | SH | BC | SBF | HA | HC |
|---|---|---|---|---|---|---|---|---|---|
| minimum-only | 57.85 | 22.01 | 65.42 | 69.22 | 67.97 | 44.80 | 66.95 | 57.30 | 57.77 |
| SAE-only | 71.22 | 21.80 | 65.46 | 69.12 | 68.15 | 43.80 | 64.91 | 57.42 | 59.54 |

## 5 CONCLUSION

This paper introduces P2RBox, the first point-supervised oriented object detector to our best knowledge. P2RBox distinguishes features through multi-instance learning, introduces a novel method for assessing proposal masks, designs a SAE Module for oriented bounding box conversion, and trains a fully supervised detector. P2RBox achieves impressive detection accuracy, with the exception of complex categories like BR. P2RBox offers a training paradigm that can be based on any proposal generator, and its generated rotated bounding box annotations can be used to train various strong supervised detectors, making it highly versatile and performance-adaptive without the need for additional parameters.

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
