## 6   APPENDIX

**Upper Limit in our method** In fact, we don't create new mask proposals; we simply choose a mask from the SAM generator using our criteria. As a result, there's a performance limit. When selecting based on IoU with the ground truth, the IoU results are displayed in Tab. 6. This result demonstrates

Table 6: IoU result of SAM (highest score), P2RBox, ceiling (always choose the highest IoU using SAE on PL and HC while others minimum).

| Method | PL | BD | BR | GTF | SV | LV | SH | TC | BC | ST | SBF | RA | HA | SP | HC | mIoU |
|---|---|---|---|---|---|---|---|---|---|---|---|---|---|---|---|---|
| SAM | 55.70 | 60.72 | 17.85 | 62.65 | 63.79 | 65.90 | 67.06 | 78.38 | 25.54 | 57.87 | 46.12 | 48.47 | 52.26 | 60.20 | 56.04 | 54.57 |
| P2RBox | 71.22 | 66.10 | 22.01 | 64.83 | 65.42 | 69.22 | 67.97 | 80.70 | 44.80 | 58.49 | 66.95 | 52.22 | 57.30 | 63.50 | 59.54 | 60.68 |
| IoU-highest | 74.08 | 70.39 | 26.23 | 78.53 | 69.61 | 73.48 | 74.91 | 83.43 | 47.14 | 64.61 | 70.08 | 58.37 | 66.51 | 66.81 | 64.11 | 65.89 |

that we have outperformed the SAM model in every category compared to simply selecting the highest score. It also highlights that for some categories, the performance remains poor due to very low upper limits, despite significant improvements from the baseline.

**Details when using Symmetry Axis Estimation Module.** Tab. 7 provides detailed information. The SAE method shows a slight decrease in IoU for some categories, which is negligible. However, it experiences a significant drop in the BD category. The issue arises because the annotation or

Table 7: Different mask2rbox method IoU results.

| Method | PL | BD | BR | GTF | SV | LV | SH | TC | BC | ST | SBF | RA | HA | SP | HC | mIoU |
|---|---|---|---|---|---|---|---|---|---|---|---|---|---|---|---|---|
| minimum-only | 57.85 | 66.10 | 22.01 | 64.83 | 65.42 | 69.22 | 67.97 | 80.70 | 44.80 | 58.49 | 66.95 | 52.22 | 57.30 | 63.50 | 57.77 | 59.68 |
| SAE-only | 71.22 | 58.14 | 21.80 | 64.94 | 65.46 | 69.12 | 68.15 | 80.37 | 43.80 | 56.00 | 64.91 | 52.87 | 57.42 | 62.95 | 59.54 | 59.78 |

ground truth for BD does not align with its symmetry axis, even when a symmetry axis is present, as illustrated in Fig. 4.

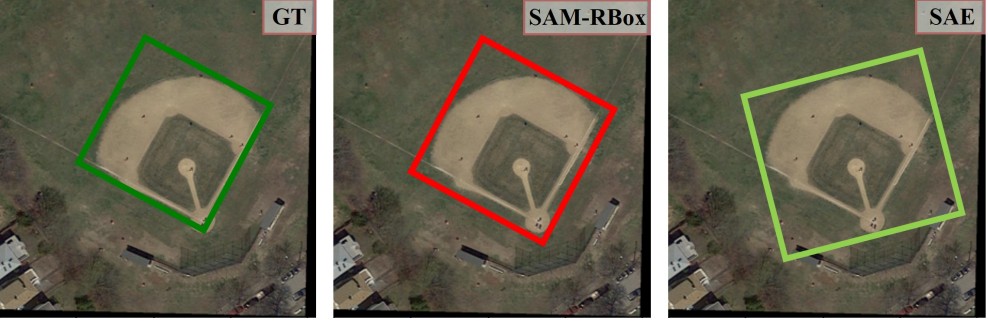

Figure 4: GT, minimum and SAE on category BD.

**The limitations on the upper performance bound for the Bridges.** category are quite restrictive. This is primarily attributed to the distinctive nature of its definition, which deviates from the conventional object definitions. In the case of bridges, they are defined as road segments that span across bodies of water, leading to a situation where there are insufficient discernible pixel variations between the left and right ends of the bridge. Consequently, this characteristic significantly hampers the performance of the SAM model. As a result, it imposes a notable constraint on the potential performance within this category. This challenge is further exemplified in Fig. 5.

**Details in Inspector Module.** We designed the coefficients of the Inspector Module to address challenges posed by small-scale objects. In cases where a small-scale object assimilates excessive background context, the increment in the denominator term $Radius$ within the coefficient formulation leads to a reduction in the $S_{offset}$. Consequently, as shown in Fig. 6 this deviation in $S_{offset}$ guides the bias observed in our selection of outcomes.

**What Result in Bad Bases using minimum bounding rectangle.** To illustrate this without loss of generality, let's consider an object that exhibits symmetry about the y-axis (see Fig. 7). We'll denote three points on the oriented circumscribed bounding box as $a$, $b$, and $c$, respectively, and their corresponding mirror points as $\hat{a}$, $\hat{b}$, and $\hat{c}$.

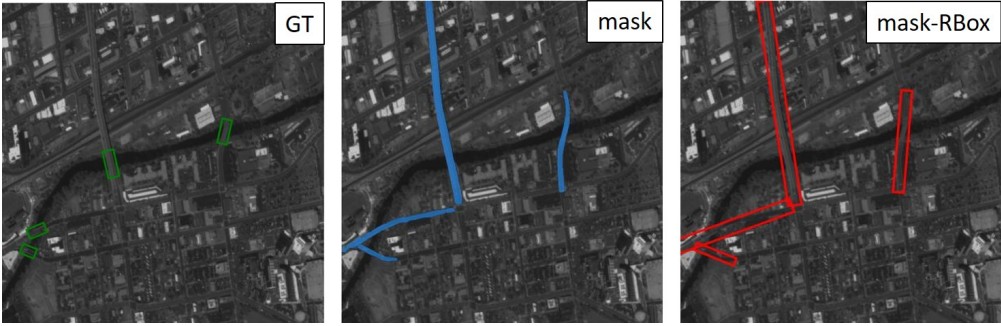

Figure 5: Category BR, with mask proposals generated by SAM with annotated point and its circumscribed rbox.

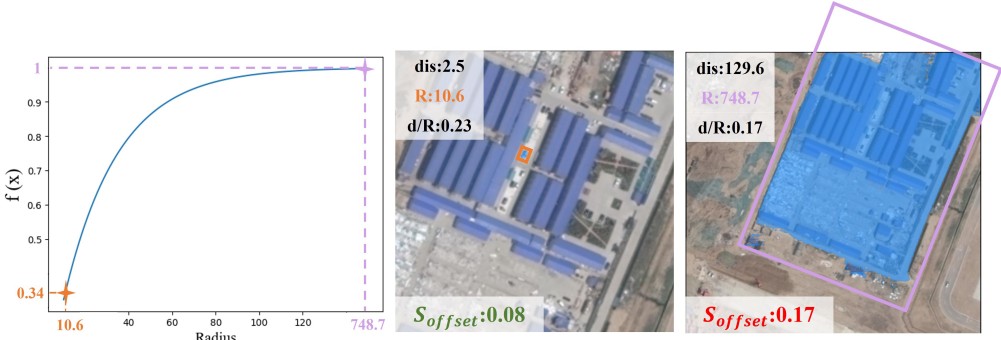

Figure 6: The influence of the coefficients of the Inspector Module.

Now, suppose there exists a minimum circumscribed bounding box, denoted as $Mbox$, which is distinct from $Rbox$. By virtue of symmetry, $Mbox$ must also exhibit symmetry, compelling its shape to be square with its diagonal aligned along the axis of symmetry. To encompass the entire object, $Mbox$ must enclose points $a$, $b$, and $c$ as illustrated in Fig. 7 (a). Let $d$ represent the length of the diagonal of $Mbox$. We have the following conditions:

$$d >= \max(h + x_a, w/2 + y_b) - \min(-x_c, y_b - w/2), \quad (19)$$

$$d^2 <= 2 \times w \cdot h. \quad (20)$$

The second inequality is derived based on the area requirement. For the more general case, as shown in Fig. 7 (b). By finding two tangent lines with fixed slopes (1 and -1), where $\alpha \cdot h$ is the distance between the intersections of these lines with the right green edge, we obtain an equation regarding the length of the diagonal:

$$d = w + \alpha \cdot h. \quad (21)$$

Specifically, if the width is equal to the height, $w = h$, the inequality simplifies to:

$$\alpha <= \sqrt{2} - 1. \quad (22)$$

In conclusion, taking an airplane as an example, as shown in the last column of Fig. 1, due to the intersection ratio $\alpha < \sqrt{2} - 1$, ambiguity arises between the minimum bounding rectangle and the oriented bounding rectangle, which is well addressed by Symmetry Axis Estimation Module.

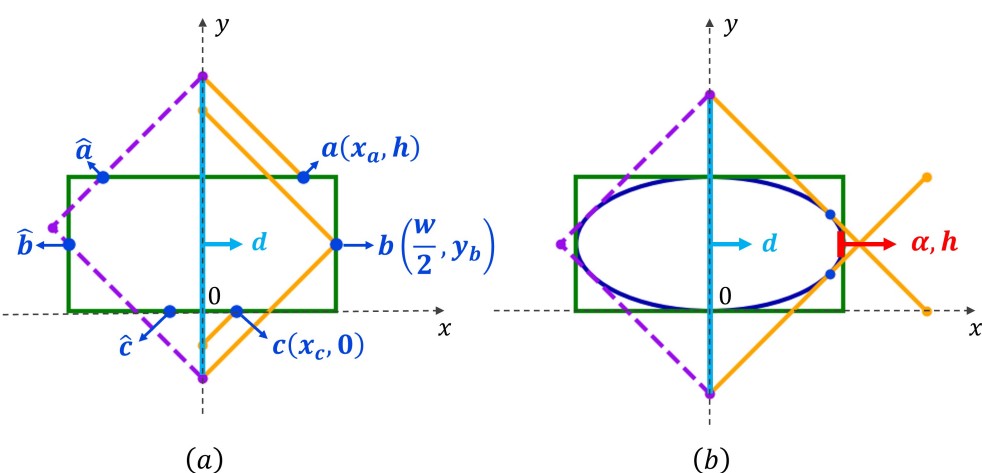

Figure 7: Convex polygon example compared to the general case.