# OpenReview forum: "P2RBOX:A SINGLE POINT IS ALL YOU NEED TRAINING ORIENTED OBJECT DETECTOR"
_ICLR.cc/2024/Conference — Submitted to ICLR 2024_

### Official Review · Reviewer_KHmW · 2023-10-14

**Soundness:** 2 fair
**Presentation:** 2 fair
**Contribution:** 1 poor
**Rating:** 3
**Confidence:** 4

**Summary:**

The authors introduce a novel framework P2RBOX for oriented object detection under point supervision. P2RBOX consists of mask generator, constrainer module and inspector module. The authors made reasonable designs for these modules and finally achieved good performance on mainstream datasets.

**Strengths:**

The motivation is interesting, and the proposed method is theoretically feasible. Experimental results show that the framework achieves good performance.

**Weaknesses:**

The content of Fig. 2 is not enough to intuitively understand the process of the proposed framework. It is suggested that the author include more details and explanations in Fig.2.

**Questions:**

The proposed framework is not innovative enough and is not a completely point-based supervision method.

---

> ### Author Response · Authors · 2023-11-21
>
> >W1: The content of Fig. 2 is not enough to intuitively understand the process of the proposed framework. It is suggested that the author include more details and explanations in Fig.2.
>
> Thank you for your feedback. I appreciate your suggestion to enhance the clarity of our training process overview (Fig. 2). We will incorporate additional visual aids and improve the representation of the point selection in the Point Bag Generation module for better comprehension.
> >Q1: Oriented object detection is a generalized form of general object detection. Please discuss the effect of the proposed framework on horizontal object detection.
>
> Thank you for your comments. haven't conducted experiments on widely-used detection datasets like COCO. But in my analysis, the final performance mainly depends on the performance of the SAM model as an anchor generator. The method proposed in this article determines a new criterion to select a more matching mask. For a general target detection data set, SAM may perform better than the DOTA data set. At the same time, the SAE module will become useless. When applied to a more diverse dataset like COCO, especially for categories like humans with challenges such as occlusion and varying poses, the boost from semantic scores might not be as pronounced. Since I have not done experiments on COCO, I will be happy to show the experimental results on COCO in the future.

---

### Official Review · Reviewer_4XWe · 2023-10-30

**Soundness:** 2 fair
**Presentation:** 2 fair
**Contribution:** 2 fair
**Rating:** 3
**Confidence:** 5

**Summary:**

This paper focuses on  point-supervised oriented object detection task, and presnts the P2RBox network. The method first leverages the SAM-based mask generator to produce the mask proposals based on single point, and then  presents Constrainer and Inspector modules to select the high-quality mask proposal. The Symmetry Axis Estimation (SAE) module are further proposed to better  generate the fitting rotated box for some categories. Based on the generated rotated bounding box, a fully-supervised method  can be trained. The extensive experiments are conducted to demstrate the effectiveness of the proposed method on DOTA.

**Strengths:**

1. The point-supervised oriented object detection is a challenging task with only point-based information. Well, weakly-spervised oriented object task is also an interesting research topic.

2. The proposed method can benefit the high-quality mask selection based on point prompt  for current zero-shot mask generator like SAM for aerial image aera.

**Weaknesses:**

1. In this paper,  some unreasonable aspects exist:
(1) Strictly speaking, this method is not a pure point-supervised oriented object detection approach. This method depends on mask proposals from SAM-based mask generator, which has been trained with lots of data.
(2)  Oriented object detection is a compromise solution between the hbox-based object detection and fine-grained segmentation tasks. This paper adopts the  "point-to-mask-to-rbox" manner. Compared with rbox with orientation information,  mask has fine-grained pose information. The approach  designs to generate the relatively coarse rbox based on the selected masks, which may put the cart before the horse.  The proposed method can solve the high-quality mask selection based on point prompt  for general mask generator like SAM for aerial image filed, which is applicable.
(3)This method follows the assumption that the annotated point of an object is close proximity to the center of the mask.  This assumption  is also not rigorous enough.
2. The proposed Symmetry Axis Estimation (SAE) module is not general for all categories, which is effective for PL and HC for DOTA dataset. For most  of categories, the peformance is descreased form the Table 7 in Appendix.  For other datasets, how to determine whether to use this module.
3. Some experimental settings is not clear, such as all models adopts r-50 model with 1x trianing schedul? Some reported results miss  the mAP results in some ablation studies, why only report mIoU?

**Questions:**

Please see the Weaknesses.

---

> ### Author Response · Authors · 2023-11-21
>
> > W1.1:(1) Strictly speaking, this method is not a pure point-supervised oriented object detection approach. This method depends on mask proposals from SAM-based mask generator, which has been trained with lots of data.
>
> You're correct in saying that our method doesn't depend only on one specific type of guidance, but it's not strongly supervised either. I want to highlight that I didn't train SAM model's details for the DOTA dataset; I used the settings given by the official website. The comparison baseline is also created using SAM. I truly value your insights on this issue.
> > W1.2:(2) Oriented object detection is a compromise solution between the hbox-based object detection and fine-grained segmentation tasks. This paper adopts the "point-to-mask-to-rbox" manner. Compared with rbox with orientation information, mask has fine-grained pose information. The approach designs to generate the relatively coarse rbox based on the selected masks, which may put the cart before the horse. The proposed method can solve the high-quality mask selection based on point prompt for general mask generator like SAM for aerial image filed, which is applicable.
>
> Thanks to the reviewers for their feedback. I understand your point of view, that is, if you already have mask-level annotation, you may not need to pay attention to the rotation box-level annotation, because the latter is relatively rough. However, I would like to highlight application-specific aspects of our method for rotating box annotation. In some scenarios, rotating box annotation still has unique advantages, such as in applications that deal with tilted targets, directional detection, or where the target direction needs to be considered. In these cases, rotating box annotations may provide more accurate information rather than just being redundant as mask-level annotations. I hope this will make you reconsider this article.
> > W1.3:(3)This method follows the assumption that the annotated point of an object is close proximity to the center of the mask. This assumption is also not rigorous enough.
>
> This is to refer to the following two articles to obtain some priors for point annotation[1][2]. Therefore, the article makes an assumption that the labeling points often present an approximate Gaussian distribution at the center of the object.
>
> > W2: The proposed Symmetry Axis Estimation (SAE) module is not general for all categories, which is effective for PL and HC for DOTA dataset. For most of categories, the peformance is descreased form the Table 7 in Appendix. For other datasets, how to determine whether to use this module.
>
> Thank you for bringing up important concerns about how well the Symmetry Axis Estimation (SAE) module works in different situations. Let me address a few points: first, it's crucial to mention that a decrease in performance when using SAE doesn't happen a lot across most categories. Also, we've explained in detail why there's a significant drop in performance for category BD in the additional information section. Additionally, the small decrease in mIoU doesn't strongly affect either the training results (measured by mAP). When deciding whether to use the SAE module, it basicly depends on whether the object is symmetric or not, which needs prior knowledge.
> > W3: Some experimental settings is not clear, such as all models adopts r-50 model with 1x trianing schedul? Some reported results miss the mAP results in some ablation studies, why only report mIoU?
>
> Thank you for giving us feedback on how we set up our experiment. I used the ResNet-50 architecture for all our models and trained them with a 1x schedule. I choose to report mIoU instead of mAP because, in my observation, mAP values for each category can vary a lot due to inaccurate label in the same training condition. Using mIoU seemed like a better way to show the differences accurately in our situation.
>
> * [1] Point-to-Box Network for Accurate Object Detection via Single Point Supervision
> * [2] AttentionShift: Iteratively Estimated Part-Based Attention Map for Pointly Supervised Instance Segmentation

---

### Official Review · Reviewer_hRnG · 2023-10-30

**Soundness:** 3 good
**Presentation:** 3 good
**Contribution:** 4 excellent
**Rating:** 6
**Confidence:** 4

**Summary:**

The first attempt is proposed at training an oriented object detector with point supervision. P2RBox achieves impressive detection accuracy, with the exception of complex categories like BR. P2RBox offers a training paradigm that can be based on any proposal generator, and its generated rotated bounding box annotations can be used to train various strong supervised detectors, making it highly versatile and performance-adaptive without the need for additional parameters.

**Strengths:**

1. The author ingeniously design the p2rbox to establish a seamless connection between point annotations and rotated boxes through the generation, constraint, and inspection of mask proposals.
2. This is an interesting attempt to train an oriented object detector with point supervision, reducing dependence on data annotation.
3. The SAE module was designed based on the spectral theorem of symmetric matrices, utilizing prior information of the object to improve the effectiveness of rotating target detection.
4. The author introduce a semantic score for the masks, enhancing the quality of the selected mask proposals from the mask generator.

**Weaknesses:**

1. Generator (SAM) is a high-performance segmentation method, and its introduction may not be fair to other methods.
2. Table 1 should include some recent SOTA methods (such as GWD[1], KLD[2], CGCDet[3], OSKDet[4], KFIoU[5],). Although the proposed p2rbox has significantly improved compared to baseline, there is still a significant gap in its performance compared to supervised methods. Is its performance persuasive enough?
3. Compared to supervised methods, the method adopts a segmentation mask generator. Will its speed be significantly affected? Please provide an analysis.

[1] Rethinking rotated object detection with gaussian wasserstein distance loss. ICML 2021.
[2] Learning high-precision bounding box for rotated object detection via kullback-leibler divergence. NeurIPS 2021.
[3] Learning Oriented Object Detection via Naive Geometric Computing. TNNLS 2023.
[4] OSKDet: Orientation-sensitive Keypoint Localization for Rotated Object Detection. CVPR 2022
[5] The KFIoU Loss for Rotated Object Detection. ICLR 2023.

**Questions:**

1. The Constrainer Module selects high-quality masks to create the union mask. Is it better to choose an optimal mask, or to generate the optimal mask through the mask generated by SAM?
2. By using SAM to segment the target, the quality of the segmentation mask obtained is relatively high. By filtering false positives through the Constrainer Module, higher performance should be achieved. However, compared to supervised methods, the performance difference is still significant. What is the main reason for the performance loss?

---

> ### Author Response · Authors · 2023-11-21
>
> > W1: Generator (SAM) is a high-performance segmentation method, and its introduction may not be fair to other methods.
>
> Your observation regarding the high-performance segmentation method SAM is valid, and we appreciate your concern. SAM makes this article not a purely weakly supervised work, and it is unreasonable to compare its performance with weakly supervised methods. Thus, our baselines are also built upon SAM and we directly use SAM with given weights without any change.
> >W2: Table 1 should include some recent SOTA methods (such as GWD[1], KLD[2], CGCDet[3], OSKDet[4], KFIoU[5],). Although the proposed p2rbox has significantly improved compared to baseline, there is still a significant gap in its performance compared to supervised methods. Is its performance persuasive enough?
>
> >Q2: By using SAM to segment the target, the quality of the segmentation mask obtained is relatively high. By filtering false positives through the Constrainer Module, higher performance should be achieved. However, compared to supervised methods, the performance difference is still significant. What is the main reason for the performance loss?
>
> Our baseline have shown the performance of SAM with only one point on DOTA. Moreover, we have conducted experiments to explore the upper limit of P2RB. Selecting the highest IoU Rbox from three different masks and training on RetinaNet yields a performance of 58.94 mAP.In contrast, our achieved result of 55.50 (selected based on our criterion) outperforms the result of 47.91 (selected based on the highest score given by SAM), showcasing substantial improvement under the same settings. The gap mainly from performance of SAM, result of not receiving specialized training in DOTA. In the future, I hope to find some inductive approach to breaking through the upper bound.
> >W3: Compared to supervised methods, the method adopts a segmentation mask generator. Will its speed be significantly affected? Please provide an analysis.
>
> Thank you for your feedback. The training speed may be affected a little. For an image with given annotated points, the “get mask” step(generate masks with given annotated points and image) cost within one second. Moreover, this article will eventually use the results of P2RBox to train strong supervised detectors. In practical applications, only strong supervision detectors are involved.
> >Q1: The Constrainer Module selects high-quality masks to create the union mask. Is it better to choose an optimal mask, or to generate the optimal mask through the mask generated by SAM?
>
> Yep, through choosing the optimal mask based on the IoU with ground truth, the performance will get to its upper limit. During training, the only information we get is a single point of every object. Our purpose is to select the best performance mask using our design criterion.

---

> ### Comment · Reviewer_hRnG · 2023-11-23
> **Comments after rebuttal**
>
> After carefully reading the author's response and comments from other reviewers, though I think this paper proposes an interesting attempt on oriented object detection, I agree this paper should be further improved. Thus, I would update my rating from "7: accept" to "6: marginally above the acceptance threshold". Thanks.

---

### Official Review · Reviewer_JsqB · 2023-10-30

**Soundness:** 3 good
**Presentation:** 3 good
**Contribution:** 2 fair
**Rating:** 5
**Confidence:** 4

**Summary:**

The P2RBox is a method that connects point annotations and rotated boxes for object detection. It uses mask proposals generated from annotated points, refines the masks with a Constrainer Module, and captures semantic nuances with an Inspector Module. The SAE module facilitates the annotation transformation. The main contribution lies in the design of more loss functions to align spatial positions.

**Strengths:**

+ This paper combines the ability of single point weakly supervised target detection and interactive large model to construct a new process of rotating target detection and obtain convincing results.

+ The mathematical representation in this paper is very clear, showing an understanding of the definition of spatial offset, positive and negative sample extraction, and various evaluation criteria.

**Weaknesses:**

+ To study target detection methods, it is important to design a novel network structure. However, most of the structures in this paper are borrowed from Oriented-RCNN, and there is no discussion on the performance/computational gain of the structure introduced by Oriented-RCNN, so it is difficult to evaluate the effectiveness of the proposed methods.


+  Extracting masks from SAM itself can be a labor-intensive task. Additionally, this approach may not scale well to larger datasets beyond just rotating target datasets like DOTA. From the standpoint of SAE, it is unclear if the axis of symmetry problem is statistically characteristic. The explanation regarding whether eigenvalue decomposition represents the statistical characteristics of the symmetry axis problem still needs further clarification.


+  Most of the loss functions proposed in this paper are fine-tuned or error-corrected based on spatial relations, without essentially changing the output representation to introduce new loss functions. For example, the centroid offset penalty of the constraint module is not actually a novel contribution point. It is only a penalty item after considering the spatial relationship. It is not recommended to describe it as a separate chapter.

**Questions:**

The part of the SAE discussion is very interesting, but does it ignore the proportion of axisymmetric targets? What is the proportion of axisymmetric targets that cause problems?  How do you consider other special situations, such as circular targets, playgrounds, baseball fields, etc?

The experimental part of this article is actually weak. This work proposes a lot of loss functions, but the experiments didn't perform the ablation study on these loss functions.  Besides, it lacks visual comparisons with other methods and lacks the analysis of failure cases.

---

> ### Author Response · Authors · 2023-11-21
>
> > W1: To study target detection methods, it is important to design a novel network structure. However, most of the structures in this paper are borrowed from Oriented-RCNN, and there is no discussion on the performance/computational gain of the structure introduced by Oriented-RCNN, so it is difficult to evaluate the effectiveness of the proposed methods.
>
> P2RBox is the first step of two stage method in this article(generate rotated box from point and train with a fully supervised detector). P2RBox is a point-to-mask network to generate rotated box as pseudo label (1st stage). Then the generated pseudo label is used to train a fully supervised detector. The performance of the supervised detector is used to measure the quality of generated pseudo label. The performance of P2RBox combined with different fully supervised detectors will not be compared together as they have different baseline to compare with. The structure of P2RB have little relationship with Oriented-RCNN, and is totally original.
>
> > W2.1: Extracting masks from SAM itself can be a labor-intensive task. Additionally, this approach may not scale well to larger datasets beyond just rotating target datasets like DOTA.
>
> Thank you for your feedback. For an image with given annotated points, the “get mask” step(generate masks with given annotated points and image) cost within one second. Moreover, this article will eventually use the results of P2RBox to train strong supervised detectors. In practical applications, only strong supervision detectors are involved.
>
> > W2.2: From the standpoint of SAE, it is unclear if the axis of symmetry problem is statistically characteristic. The explanation regarding whether eigenvalue decomposition represents the statistical characteristic.
>
> Since DOTA is a data set from an overhead perspective, occlusion problems usually do not exist, and the objects are all rigid bodies. Therefore, the problem about the symmetry direction described in the main text only exists in the PL and HC categories, and is prevalent in these two categories.
> > Q1.1: The part of the SAE discussion is very interesting, but does it ignore the proportion of axisymmetric targets?
>
> In the DOTA dataset, most object categories show symmetry, except for the HARBOR category. DOTA, being a dataset of remote sensing images, is mainly captured from a top-down view, meaning that whether the object is symmetrical or not is only related to its category excluding occlusion situations.
> > Q1.2: What is the proportion of axisymmetric targets that cause problems?
>
> The ambiguity occurs when the orientation of the minimum bounding rectangle differs from the annotation direction, particularly in the helicopter and aircraft categories. This creates a unique challenge for these categories, unlike the others.
>
> > Q1.3: How do you consider other special situations, such as circular targets, playgrounds, baseball fields, etc?
>
> The SAE module is versatile across various object shapes, providing two perpendicular directions. While directions for nearly circular objects may not have clear geometric significance, it doesn't seem to affect detection performance because a rectangle represented by any two vertical direction is suitable. For other categories that is symmetric like baseball fields, ship and palygrounds, SAE module’s outputs are same with minimum bounding boxes, showing its direction.
>
> > Q2: The experimental part of this article is actually weak. This work proposes a lot of loss functions, but the experiments didn't perform the ablation study on these loss functions. Besides, it lacks visual comparisons with other methods and lacks the analysis of failure cases.
>
> Thank you for your valuable advice. I will add more experiments to ensure the generalizability of the method.

---

### Official Review · Reviewer_oNSm · 2023-10-31

**Soundness:** 2 fair
**Presentation:** 2 fair
**Contribution:** 2 fair
**Rating:** 3
**Confidence:** 5

**Summary:**

This paper proposes a P2RBox network, which leverages point annotations and a mask generator to create mask proposals, followed by filtration through Inspector Module and Constrainer Modulel. Specifically, in order to achieve point-supervised oriented object detection, the paper uses a powerful SAM model to generate initial higher-quality masks, and then refines it through several well-designed designed modules, finally the the generated masks are converted into pseudo rotated bounding box for training the oriented object detector.

**Strengths:**

- This topic is currently new and no one seems to have studied it yet.
- The issue about the masks generated by SAM, i.e the ambiguity arises between companion scores and the best-performing mask, is interesting and reflects the current problems that may need to be solved when using SAM as a tool.
- Compared with the baseline, the proposed algorithm significantly improves the performance.

**Weaknesses:**

In my opinion, the most controversial point of the article is the use of SAM.
- As we all know, SAM is a fully supervised visual model, and its training data over 1 billion masks on 11M licensed and privacy respecting images. Therefore, the method proposed in this paper does not seem to be the so-called weakly supervised oriented object detection, more like zero shot object detection. This is extremely unfair to compare with other truly weakly supervised algorithms.
- The authors used the mask generated by SAM to generate the rotated bounding box, which seems to be redundant and unnecessary. In my opinion, if a higher quality mask can be produced, there is no need to study other rough object representations, i.e. horizontalrotated/poly bounding box. Therefore, I'm more curious the performance comparison of instance segmentation models trained by masks before and after refinement on commonly used segmentation data sets (i.e. COCO, LVIS). Relevant experiments are given in SAM.
- As mentioned earlier, the issue of masks generated by SAM found by the authors is interesting, but the proposed solution seems to be engineered and customized. The large number of hyperparameters and thresholds/scores makes me question the generality of the proposed method, and the authors only used one dataset (i.e. DOTA) for verification in the paper, which is not credible.
- The title is too exaggerated (i.e. single point is all you need). According to the author's experimental results, even if a powerful SAM is used, the performance is still far behind the fully supervised algorithm, e.g. RetinaNet vs. P2RBox (RetinaNet): 67.83% vs. 55.50% (-12.33%).
- Based on the above analysis, I think the name of the method is more suitable to be called SAM2RBox.

In general, the method proposed (i.e. SAM based) in the paper does not completely match the topic (i.e. weakly supervised object detection) it is trying to solve, resulting in an unsatisfactory discussion. Moreover, the customized method design and insufficient experiments make me think that this article is not suitable for publication yet, and I temporarily reject it.

**Questions:**

- The article lacks details on how point annotations are generated.
- The authors need to rethink the story of the proposed method, i.e this is not a simple weakly supervised detection method. It seems more appropriate to call it zero shot or a combination of the two?
- The writing of the article needs to be better improved, especially the organization of formula symbols.
- Experiments on more dataset are needed, especially the ablation experiments of hyperparameters. It is better to use datasets from different scenarios and not just remote sensing datasets.
- Appropriate failure cases need to be discussed to analyze why the current performance is still far below the performance of fully supervised algorithms.
- For Figure 3, I would rather recommend a visual comparison of other similar methods, e.g. P2RBox-H2RBox and SAM (xxx) in Table 1.
- Please place the table or figure at the top of each page.

---

> ### Author Response · Authors · 2023-11-21
>
> > W1: As we all know, SAM is a fully supervised visual model, and its training data over 1 billion masks on 11M licensed and privacy respecting images. Therefore, the method proposed in this paper does not seem to be the so-called weakly supervised oriented object detection, more like zero shot object detection. This is extremely unfair to compare with other truly weakly supervised algorithms.
>
> The use of the SAM model makes it no longer a purely weakly supervised task, but the baseline in the article is also designed based on SAM. I hope this will make you reconsider this article.
>
> > W2: The authors used the mask generated by SAM to generate the rotated bounding box, which seems to be redundant and unnecessary. In my opinion, if a higher quality mask can be produced, there is no need to study other rough object representations, i.e. horizontalrotated/poly bounding box. Therefore, I'm more curious the performance comparison of instance segmentation models trained by masks before and after refinement on commonly used segmentation data sets (i.e. COCO, LVIS). Relevant experiments are given in SAM.
>
> Thanks to the reviewers for their feedback. I understand your point of view, that is, if you already have mask-level annotation, you may not need to pay attention to the rotation box-level annotation, because the latter is relatively rough. However, I would like to highlight application-specific aspects of our method for rotating box annotation. In some scenarios, rotating box annotation still has unique advantages, such as in applications that deal with tilted targets, directional detection, or where the target direction needs to be considered. In these cases, rotating box annotations may provide more accurate information rather than just being redundant as mask-level annotations. We will further consider how to explicitly highlight the unique features of our approach and adapt accordingly in the paper. Thank you again for your helpful suggestions, we will take them seriously and make improvements accordingly.
>
> >W3: As mentioned earlier, the issue of masks generated by SAM found by the authors is interesting, but the proposed solution seems to be engineered and customized. The large number of hyperparameters and thresholds/scores makes me question the generality of the proposed method, and the authors only used one dataset (i.e. DOTA) for verification in the paper, which is not credible.
>
> Thank you for your feedback. The problem of large number of parameters in this article is indeed a shortcoming. As an author, I will add more experiments to prove that my experimental settings are actually relatively general and not specially designed.
>
> > W4: The title is too exaggerated (i.e. single point is all you need). According to the author's experimental results, even if a powerful SAM is used, the performance is still far behind the fully supervised algorithm, e.g. RetinaNet vs. P2RBox (RetinaNet): 67.83% vs. 55.50% (-12.33%).
>
> Thank you for your reply. I'm sorry that the title of this article is indeed exaggerated and I will change it. I hope it does not affect the contribution and innovation of this article. I hope this will make you reconsider this article.
>
> > Q1: The article lacks details on how point annotations are generated.
>
> The generation of point annotation is not the mean point of this article. In fact, the point annotations are generated through center point with slightly adjustment [1][2] to ensure it always locate on the object even for object of shape L.
>
> > Q5: Appropriate failure cases need to be discussed to analyze why the current performance is still far below the performance of fully supervised algorithms.
>
> Thanks for your advice. I will add more analysis experiments in the article, including parameter ablation and results on more data sets. I will also refer to your valuable opinions to add contrast in the visualization part.
>
> * [1] Point-to-Box Network for Accurate Object Detection via Single Point Supervision
> * [2] AttentionShift: Iteratively Estimated Part-Based Attention Map for Pointly Supervised Instance Segmentation

---

> ### Comment · Reviewer_oNSm · 2023-11-23
>
> After reading the comments of other reviewers, I find that many comments are consistent, and some of the comments are also acknowledged by the authors. Due to the extensive revisions needed to this paper and the lack of substantial feedback from the authors, I think the current version is not suitable for publication. Therefore, I maintain my original rating.

---

### Official Review · Reviewer_a8aF · 2023-10-31

**Soundness:** 3 good
**Presentation:** 3 good
**Contribution:** 3 good
**Rating:** 3
**Confidence:** 3

**Summary:**

Oriented object detection excels in identifying objects of any orientation, while point annotation provides a cost-effective method but lacks size and orientation data. The study introduces the P2RBox network that uses point annotations and a mask generator to produce mask proposals, which are then refined and transformed into rotated box annotations for training advanced detectors. With the integration of the Oriented R-CNN, the P2RBox achieves notable performance, marking the first attempt to train an oriented object detector using point supervision.

**Strengths:**

1.	Using points to generate oriented bounding box is interesting.
2.	This paper proposed SAE module to generate the oriented bounding box from masks.

**Weaknesses:**

1.	The MIL is widely used in weakly supervised object detection[1] and point-based object detection[2].
2.	This paper used SAM to generate the mask proposals. However, the training of SAM used many complete mask annotations. Therefore, the proposed method still rely on the complete mask supervision, which is not suitable to be considered as a point-supervised method. It is also not consistent with the tile “A SINGLE POINT IS ALL YOU NEED.”
3.	The proposed SAE only significantly outperforms minimum in PL category. In some other cases, it is even lower than minimum.
4. Can the authors compare the proposed method with point supervised instance segmention[3], in the same supervision setting?

Minor:
1.	“In many cases, the annotated point of an object is typically positioned in close proximity to the center of the mask (Chen et al., 2022). ” “close proximity” is a redundancy. Proximity means closeness, nearness; therefore “close proximity” means “close closeness” or “near nearness.”
[1] Bilen, Hakan, and Andrea Vedaldi. "Weakly supervised deep detection networks." Proceedings of the IEEE conference on computer vision and pattern recognition. 2016.
[2] Papadopoulos, Dim P., et al. "Training object class detectors with click supervision." Proceedings of the IEEE Conference on Computer Vision and Pattern Recognition. 2017.
[3] Bowen Cheng, Omkar Parkhi, and Alexander Kirillov. Pointly-supervised instance segmentation. In Proceedings of the IEEE/CVF Conference on Computer Vision and Pattern Recognition, pp. 2617–2626, 2022

**Questions:**

See the weakness. My main concern is the usage of SAM, which makes this method not suitable to be considered as a pure point-supervised oriented object detector.

---

> ### Author Response · Authors · 2023-11-21
>
> > W1: The MIL is widely used in weakly supervised object detection[1] and point-based object detection[2].
>
> A1: While MIL finds extensive application in the field of weak supervision, in specific situation, how to define and design the Bag to provide a valid supervision, guiding the model to optimize in the right direction is the crucial part. The key contribution of this article lies in the design of point bag construction, which differs from previous weak supervision tasks.
> > W2: This paper used SAM to generate the mask proposals. However, the training of SAM used many complete mask annotations. Therefore, the proposed method still rely on the complete mask supervision, which is not suitable to be considered as a point-supervised method. It is also not consistent with the tile “A SINGLE POINT IS ALL YOU NEED.”
>
> A2: Thank you for your thoughtful feedback. Regarding my title, it's really inappropriate. The use of the SAM model makes it no longer a purely weakly supervised task, but the baseline in the article is also designed based on SAM. I hope this will make you reconsider this article.
> > W3: The proposed SAE only significantly outperforms minimum in PL category. In some other cases, it is even lower than minimum.
>
> A3: Thank you for your comment. The purpose of the design of the SAE module is to solve the problem of inconsistent orientations of the minimum circumscribed rectangle of the symmetry target. This problem is currently only observed on aircraft and helicopter categories. For other categories, the applicability of asymmetric targets is not as good as the minimum circumscribed rectangle, such as the SP category (mini 63.5 vs SAE 62.95), but the difference is not significant. For other symmetry categories, such as SH (mini 67.97 vs SAE 68.15), LV (mini 69.22 vs SAE 69.12), BC (mini 44.80 vs SAE 43.80), the overall positive or negative IoU impact is not certain. Through some visualization, when SAM cannot give an accurate mask or the object is occluded, the impact of different decisions (mini or SAE) on IoU becomes unpredictable. To sum up, SAE is only suitable for the generation of directional frames with symmetrical targets as the direction.
> > W4: Can the authors compare the proposed method with point supervised instance segmention[3], in the same supervision setting?
>
> A4: Thank you for your comment. Adding a point-supervised instance segmentation as a reference experiment is indeed a solution that can be used as a baseline. However, the reference you gave is based on multi-point supervision rather than a segmentation work based on single-point supervision. I think the following work may meet the requirements of single-point supervision: AttentionShift: Iteratively Estimated Part-Based Attention Map for Pointly Supervised Instance Segmentation. Since there is no mask-level annotation on DOTA, training this model is not yet possible.

---

> ### Comment · Reviewer_a8aF · 2023-11-22
>
> I have read the responses and the comments of other reviewers. I decided to keep my original rating.

---

### Meta-Review · Area_Chair_GGzD · 2023-12-03

**Metareview:**

The paper introduces P2RBox, a method for oriented object detection using point annotations. It addresses the challenge of efficiently detecting objects with arbitrary orientations by leveraging cost-effective point annotations instead of traditional bounding boxes.

Reviewer hRnG provided a somewhat positive review, acknowledging the novelty of the proposed method and its good results. However, concerns were raised, aligning with other reviewers, about using SAM as a mask proposal method, potentially categorizing the proposed method as a zero-shot detection rather than a weakly supervised oriented object detection. The reviewer also requested comparisons with state-of-the-art detectors. Despite the authors' rebuttal, the reviewer thinks that the paper need further work and consequently lowered the rating.

Five out of six reviewers expressed disfavor toward the paper. Their main concern centered on the use of SAM as a mask proposal method, which might categorize the proposed method as a zero-shot detector rather than a weakly supervised oriented object detector. None of the reviewers altered their rating based on the authors' responses.

Considering the discussions and the authors' limited response during the rebuttal, I am unable to recommend acceptance. While the proposed method is interesting, it appears to function more as a zero-shot detector rather than a weakly supervised oriented object detector. The paper requires additional time and effort to delve deeper into this concept.

**Justification For Why Not Higher Score:**

While the proposed method is interesting, it appears to function more as a zero-shot detector rather than a weakly supervised oriented object detector. The paper requires additional time and effort to delve deeper into this concept.

**Justification For Why Not Lower Score:**

N/A

---

### Decision · Program_Chairs · 2024-01-16

Reject